# A New Statistical Features Based Approach for Bearing Fault Diagnosis Using Vibration Signals

**DOI:** 10.3390/s22052012

**Published:** 2022-03-04

**Authors:** Muhammad Altaf, Tallha Akram, Muhammad Attique Khan, Muhammad Iqbal, M Munawwar Iqbal Ch, Ching-Hsien Hsu

**Affiliations:** 1Department of Electrical and Computer Engineering, COMSATS University Islamabad, Wah 47000, Pakistan; mohammadaltaf@gmail.com (M.A.); tallha@ciitwah.edu.pk (T.A.); miqbal1976@gmail.com (M.I.); 2Department of Computer Sciences, HITEC University Taxila, Taxila 47080, Pakistan; attique.khan@hitecuni.edu.pk; 3Institute of Information Technology, Quaid-i-Azam University, Islamabad 44000, Pakistan; mmic@qau.edu.pk; 4Department of Computer Science and Information Engineering, Asia University, Taichung 400-439, Taiwan; 5Department of Medical Research, China Medical University Hospital, China Medical University, Taichung 400-439, Taiwan; 6Guangdong-Hong Kong-Macao Joint Laboratory for Intelligent Micro-Nano Optoelectronic Technology, School of Mathematics and Big Data, Foshan University, Foshan 528000, China

**Keywords:** vibration signal analysis, condition based maintenance, time domain analysis, frequency domain analysis, machine learning, classification

## Abstract

In condition based maintenance, different signal processing techniques are used to sense the faults through the vibration and acoustic emission signals, received from the machinery. These signal processing approaches mostly utilise time, frequency, and time-frequency domain analysis. The features obtained are later integrated with the different machine learning techniques to classify the faults into different categories. In this work, different statistical features of vibration signals in time and frequency domains are studied for the detection and localisation of faults in the roller bearings. These are later classified into healthy, outer race fault, inner race fault, and ball fault classes. The statistical features including skewness, kurtosis, average and root mean square values of time domain vibration signals are considered. These features are extracted from the second derivative of the time domain vibration signals and power spectral density of vibration signals. The vibration signal is also converted to the frequency domain and the same features are extracted. All three feature sets are concatenated, creating the time, frequency and spectral power domain feature vectors. These feature vectors are finally fed into the K- nearest neighbour, support vector machine and kernel linear discriminant analysis for the detection and classification of bearing faults. With the proposed method, the reduction percentage of more than 95% percent is achieved, which not only reduces the computational burden but also the classification time. Simulation results show that the signals are classified to achieve an average accuracy of 99.13% using KLDA and 96.64% using KNN classifiers. The results are also compared with the empirical mode decomposition (EMD) features and Fourier transform features without extracting any statistical information, which are two of the most widely used approaches in the literature. To gain a certain level of confidence in the classification results, a detailed statistical analysis is also provided.

## 1. Introduction

Rotating machinery using bearings is one of the most important components of a wide range of mechanical setups from small motors to turbines, compressors and heavy ground and air vehicles [1,2]. Different faults arise during the mechanical and industrial process, generating vibration and Acoustic Emission (AE) signals [3,4]. These signals have different characteristics due to the nature of faults, the complexity of the underlying industrial setup and the correlation between different mechanical components [5,6]. Preemptive measure and real time monitoring of these signals can avoid severe losses and disastrous failures [7,8] and hence has received considerable attention [9]. Different methods of condition based monitoring are being developed and used, including but not limited to oil debris, vibration, acoustic emission, electrostatic and temperature analysis [10,11,12,13].

Early detection and localisation of mechanical faults, such as misalignment, gear faults mass unbalance and cracks along with their propagation in rotating shafts and gear wheels are possible by analysing vibration [14] and acoustic emission [15] signals, with AE having a much wider range of frequencies as compared to vibration signal [16]. The AE signals are generally transient elastic waves resulting from fast strain energy discharge due to damage on or within the material surface [17].

Both the AE and vibration signals can effectively be used for the detection and localization of defects in rotating machinery. However, the AE signal outperforms the vibration signal in case early and preemptive detection is required and also in fault detection in low speed rotating machines due to the limited efficiency of vibration signals as compared to AE signals [10]. These signals are non-stationary in nature and are complicated to analyse due to the heavy background noise of industrial set up [18]. Therefore, state-of-the-art signal processing algorithms and pattern recognition techniques are combined to monitor these signals for the detection and classification of faults. The signal processing is used to extract different time domain, frequency domain and time-frequency domain features of these signals for use with Artificial Intelligence (AI) techniques [8,18,19].

Statistical analysis of time domain AE signals have detected shaft cracks using features like energy, duration count and average signal level [20]. A similar approach is used in [21], where the amplitude and energy of the AE signals are used for defects in roller bearing. Abdullah et al. [22] detected the bearing defects and its sizes from both the time domain AE and vibration signals, using its amplitude and the root mean square (RMS) values. Time domain and wavelet domain kurtosis of the vibration signal, along with the Largest Lyapunov Exponent (LLE), were used in [19] for the analysis of slew bearing defects. These features were fed to kernel based regression for detection of the damage and estimation of the useful life of the bearings under testing. Antoni et al. [23] used a fast kurtogram for the detection of transient faults with a similar computational complexity to that of Fast Fourier Transform (FFT). Kurtosis and its different variations, such as kurtogram, spectral kurtosis, adaptive spectral kurtosis, and Short Term Fourier Transform (STFT) based kurtosis, have been used extensively by the research community for the analysis of vibration signals from rotating machinery; interested readers are referred to [24].

Feature extraction is one of the most key factors in reducing the classification errors; however, recent developments in the field of Deep Neural Networks that have the ability to extract features from original signals are being used in fault diagnosis [25,26,27,28]. That is why deep learning is swiftly paving the way for condition based maintenance with reproducible results. Mingyong et al. used Convolutional Neural Networks (CNN) with time domain vibration signals for fault diagnosis, with 96% accuracy on Case Western Reserve University (CWRU) data, and classified the faults into rolling element, inner and outer ring faults [29]. Inner and outer race faults are detected using the root mean square measure of the vibration signal and adaptive neuro-fuzzy inference system with an accuracy of 98.37% [30]. The authors proposed a feature learning approach in [31] using vibration and current signal with deep CNN, claiming an accuracy of 98% if the same machine is used for feature learning and testing, and an accuracy of 92% if the model trained on one machine is used for testing another machine. The algorithm can also provide other information such as rotational speed and number of balls and so forth.

The Fourier transform is used for analysis of signals in frequency domain [32]; however, it lacks time information. Therefore, to provide local feature information, a time-frequency technique like STFT is used to calculate the local features using the windowing approach [33]. In the time-frequency domain, the Wavelet Transform is one of the most appropriate approaches for the detection and localisation of faults [34,35,36,37]. Other time-frequency approaches to detect abnormal conditions such as misalignment, rotor-stator rub and shaft cracks in the vibration signals, including but not limited to Continuous Wavelet Transform (CWT) and Hilbert Huang Transform (HHT). The latter is also used to effectively monitor defects in bearing [38], while the former is used for early stage fault detection in the outer race [6]. Envelop detection with autocorrelation is used to detect faulty patterns at the inception stage in low SNR (Signal to Noise Ratio) AE signals [6], with wavelet transform for de-noising of the AE signals. Spectral components of the Intrinsic Mode Functions (IMF) are used to analyse both the AE and vibration signals for the detection of bearing defects, broken bar and unbalanced load distribution in the indoor motors [39]. However, it needs the a priori knowledge of the number of modes in which the signal is required to be decomposed. Features from raw vibration signals were extracted by [8], using Empirical Mode Decomposition algorithm and CNN. These features were fed to SVM and CNN training algorithms for the classification of faults into the outer race, inner race and ball faults. In another study, Khan et al. used EMD with KNN to classify the different machine states into normal, cracking, offset pulley and wear states [40]. In [41], the EMD is combined with deep neural networks to classify the faults into roller, inner and outer races with an accuracy of 98.5%. The authors have detected and classified faults using ensemble empirical mode decomposition (EEMD) and SVM. The EMD is used to decompose the vibration signal, isolating and denoising high frequency IMFs using Pearson correlation coefficients and the wavelet semi-soft threshold, respectively. The Eigen vector of the signal is used as a feature vector for the SVM to classify faults into inner race, outer race and rolling elements, with an accuracy of 100% [42]; however, the 100% accuracy seems to be on the higher end. Delprete et al. [43] used orthogonal empirical mode decomposition analysis (a time-frequency) to detect faults in the inner and outer raceway of the bearings using vibration signals.

Informative signatures can be extracted form vibration signals in the form of wavelets transform. Morlet wavelet transform is used to extract features from vibration signals that were then used with artificial neural networks and SVM to classify the signals into ball fault, inner and outer race fault, with 95.271% accuracy for SVM and 87.25% for ANN [44]. The authors of [45] combined features like vibration severity, dyadic wavelet energy time-spectrum and coefficients power spectrum of maximum wavelet energy level and fed it to SVM for classification into a normal state, eccentric axle fault (EAF), bearing pedestal fault (BPF), and sealing ring wear fault. The SVM parameters were optimized using the modified shuffled frog-leaping algorithm, claiming a maximum accuracy of 96% [45]. A Deep Fault Diagnosis (DFD) method is proposed for rotating machinery with scarce data labels. In this procedure, discriminative STFT data are obtained from the spectrogram of the vibration signals. Several SVM models are trained with different features selected from the pool with scarce labels and the most discriminative features and the best SVM models were selected, hence forming an augmented training set. This augmented training set was then forwarded to a 2-D deep CNN. The proposed algorithm classified different faults with an accuracy of 98.4% [46].

The research community has also used image analysis for diagnosing faults; here, image sparse representation is proposed to extract meaningful features from redundant information present in images using orthogonal matching pursuit and K-singular value decomposition algorithms along with 2-D PCA. The features are used with a minimum distance to classify into the inner race, outer race and ball faults with an average accuracy of 99.92% [47]. In [48], Moussa et al. proposed an algorithm for bearing fault diagnosis based on the probability of image recognition techniques under constant and variable speed conditions using average PCA. The paper used vibration spectrum imaging of the vibration signal obtained from faulty and normal bearings with CNN for classification. In [49], the time domain vibration signal was first segmented using a time-moving segmentation window and was then transformed into a spectral image for training and testing with CNN. The proposed scheme provided good accuracy at different levels of noises and speeds. Similarly, the time domain, frequency domain and time-frequency domain parameters are used to detect faults in axial piston pumps with deep belief networks [50].

The literature review as discussed above does not cover the complete review of the topic; however, it provides quite a clear picture of the work being contributed by the research community. The statistical approach, frequency domain and time-frequency domain provide different levels of accuracy with conventional machine learning techniques and deep neural networks. In this research work, the statistical and frequency domain techniques are analysed to detect and classify the faults in rotating machinery using conventional machine learning algorithms. The vibration signal from the faulty and healthy rotating machinery was recorded. From these vibration signals, the statistical features, such as Skewness, Kurtosis, Average and root mean square (RMS) values of time domain vibration signals, are considered. The same features were then extracted from the second derivative of the time domain vibration signals, making a vector of eight features. For the second and third steps, these features were calculated by first taking the Fourier Transform and Power Spectral Density of the vibration signals. All three sets of feature vectors were concatenated creating time domain, frequency domain and spectral power domain feature vectors. SVM, KNN and KLDA classifiers are used for the classification of signals into outer race fault, inner race fault, ball fault and healthy signal. Simulation results showed that the KLDA resulted in an accuracy of 99.13% for our proposed method using PSD, followed by the statistical features with KLDA giving an accuracy of 98.275%, showing the strength of our proposed algorithms.

These results were then compared with those of EMD, which has a very good accuracy as given in the literature review discussed above. In this case, the accuracy of the EMD feature vector was 97.01% with SVM. It is important to note that the size of the feature vector for EMD is 14 × 160,000 in this particular case and that of the Statistical, FT and PSD is 8 × 228. This also shows the effectiveness of the proposed approach in terms of size of the feature vector and hence less computation.

The primary contributions of this research are enumerated below:(1)We exploit the behaviour of feature extraction based on the performance of selected classifiers;(2)We propose to utilise the statistical features including kurtosis, skewness, average and root mean square of the selected window;(3)To extract the more meaningful information, we extracted the same features from the second derivative;(4)We utilise both frequency and time domain signal information for feature extraction—employing the moving window concept;(5)We propose a system which generates an accuracy of more than 95%—utilising less than 5% of information and achieving the reduction percentage of 95%.

In the rest of the paper, signal processing techniques for extracting the proposed features and calculating FFT and PSD and so forth are discussed in Section 2. The discussion of these features for fault detection is given Section 3. The vibration signals are analysed using signal graphs of different faults. A discussion on the classifier and classification of faults into outer race, inner race, and ball faults is presented in Section 4, with the concluding remarks given in Section 5.

## 2. Feature Extraction

In this section, the digital signal processing (DSP) techniques that are used for feature extraction are discussed. These techniques can be categorised under three main headings, the statistical features of time domain signal, the statistical features of signal in Fourier domain and the statistical features of signal’s Power Spectral Density. The raw vibration data was used to extract statistical features like maximum value, minimum value, standard deviation, mean, median, variance, skewness, kurtosis, range, Fisher Information Ratio [51,52], Petrosian Fractal Dimension [51], and entropy. The results of skewness, kurtosis and standard deviation and second derivative and so forth, to name a few. The features that were selected are mean value, standard deviation, skewness, kurtosis and second derivative. These features were calculated for time domain vibration signal, for the signal in Fourier domain and for the same signal after calculating its PSD. These features were then fed to the SVM, KNN and KLDA algorithms for classification into outer race fault, inner race fault, and ball fault as given in the block diagram of Figure 1. As is shown, the raw data are extracted, pre-processed and its statistical features like skewness, kurtosis, average and RMS values are calculated. It is important to note that extracting these statistical features in the Fourier domain and from the PSD of the vibration signal is not used previously, to the best of our knowledge. This proposed method has also reduced the number of data points and hence computational requirements as discussed in Section 4. The vibration signal of healthy and faulty bearings were recorded, using vibration sensors, shaft rotating at a rate of 800 revolutions per minute and a sampling rate of 40,000 samples per second. Figure 2 shows the specifications and block diagram of the test rig with ball bearing model, fault types and specifications of the data acquisition board.

The equations for skewness, kurtosis, standard deviation and second derivative are given below. The skewness is a measure of symmetry or the lack of symmetry, and is zero for the data with normal distribution and should be zero for any symmetric data. The kurtosis is a measure of whether the data are heavy-tailed or light-tailed relative to a normal distribution. Both provide important statistical characteristics of a population. The mean and the standard deviation provide the central value; however, some times it is required to find how far the data are spread, and that is measured by variance and standard deviation. The mathematical representation of all these statistical measures is given here for ready reference. For a 1-Dimensional data $Y_{1},Y_{2},...,Y_{N},$ the mathematical representation of these statistical measures is given as: $$\mathrm{Mean}=\bar{X}=\frac{1}{n}\sum_{i=1}^{n}Y_{i}$$
$$\mathrm{Std}=\sqrt{\frac{1}{N-1}\sum_{i=1}^{N}{\left(Y_{i}-\bar{Y}\right)}^{2}}$$
$$\mathrm{Skewness}=\frac{\sum_{i=1}^{N}{\left(Y_{i}-\bar{Y}\right)}^{3}/N}{s^{3}}$$
$$\mathrm{kurtosis}=\frac{\sum_{i=1}^{N}{\left(Y_{i}-\bar{Y}\right)}^{4}/N}{s^{4}}.$$

In the above equations, the $\bar{Y}$ is the mean, *s* is the standard deviation, and *N* is the number of data points (NOTE: please see if all symbols are correctly defined here?). The equation below shows the first difference as implemented in $Matlab^{T}$. Taking the same equation twice will give an approximation of second derivative. The first order derivative gives non-zero values along the ramp while the second order derivative gives non-zero and a sign change at the onset and end of a ramp, and a much aggressive response at the spikes along with a sign change. Thus, it gives the features that can be easily differentiated. The output vector of the second difference is then used to calculate the statistical measures as given above.
Diff=[Y(2)−Y(1)Y(3)−Y(2)...Y(N)−Y(N−1)].

In a second attempt, the Fourier Transform and Power Spectral Density of the vibration signal are calculated and the statistical values of the spectral components were used as a feature vector for the classification giving the concept of spectral mean, skewness, kurtosis, standard deviation and second order difference. Fourier transform is calculated using the Fast Fourier Transform (FFT) algorithm as in Equation (Equation 1), while for PSD the Fourier Transform of the autocorrelation function is required. For in depth details the interested readers are referred to [53].
(1)$$\begin{array}{c} y\left[k\right]=\sum_{n=0}^{N-1}x\left[n\right]e^{-\frac{j2\pi nK}{N}}\;k=0,1,2,3,...N-1, \end{array}$$
where y[k] is the Fourier transform and x[n] is the signal under test.

## 3. Signal Analysis for Fault Detection

The vibration data collected from machine under test with the specifications given in Figure 2 were used for extracting different features as discussed in Section 2. The vibration signal was recorded at a sampling rate of 40,000 samples per second and was divided into chunks of one second data for calculating these features. Figure 3, Figure 4, Figure 5 and Figure 6 show candidate feature vectors using the three different set of features. Time domain features are shown in Figure 3 and it can be seen that the average and standard deviation features show clear segregation between different types of faults; however, the skewness and kurtosis cannot be easily interpreted visually. Similar behaviour can be seen in the frequency domain representation of Average, Kurtosis, Skewness and Standard Deviation as shown in Figure 4. These features are obtained by first taking the Fourier Transform of the vibration signal, dividing it into chunks of one second data. The Average, Kurtosis, Skewness and Standard Deviation of the Fourier Transform of that one second data is taken. Similarly, the PSD of one second vibration data is calculated and its Average, Kurtosis, Skewness and Standard Deviation is shown in Figure 5. Figure 6 shows the Average values of time domain, Fourier domain and of PSD of the vibration signal; however, here these values are calculated after taking the second derivative of the signal. These features are much clearer as compared to the other features given in Figure 3, Figure 4 and Figure 5d; however, to properly analyse these features and classify them into different types of faults, these features are forwarded to the SVM, KNN and KLDA. The results are discussed in the next Section.

## 4. Classification of Faults

Let $s={\left[s_{1},s_{2},...,s_{p}\right]}^{⊤}\in R^{p}$ is a *p* dimensional feature vector representing statistical and frequency domain features of the vibration signal. The training data matrix is given by $V={\left\{s_{j}\right\}}_{j=1}^{v}\in R^{p\times v}$, where *v* is the normalised feature vectors and *c* is the number of classes with the discrete class labels represented by $Y={\left\{y_{j}\right\}}_{j=1}^{v}$. The fault classification problem estimates the label $y_{t}$ of a test feature vector $s_{t}\in R^{p}$ given the labelled training data V. In order to analyse the effectiveness of our proposed features for fault classification, the feature vectors are used with classical supervised learning algorithms like Support Vector Machines (SVM), Nearest Neighbour (NN), and Kernel Linear Discriminant Analysis (KLDA).

KNN is a popular algorithm based on distance measure between two feature vectors $s_{i}$ and $s_{j}$, using the Mahalanobis distance which is defined as [54]:(2)$$d={\left(s_{i}-s_{j}\right)}^{⊤}C^{-1}\left(s_{i}-s_{j}\right),$$
where $C\in R^{p\times p}$ represents the covariance matrix obtained from training feature vectors. The C can be computed as a diagonal matrix in case of small training sample sizes, with the feature variances as the diagonal elements.

Support Vector Machine was designed as supervised learning method for two class classification problems ($y_{j}\in \left\{1,-1\right\}$), but was subsequently developed for multiple class approach. In this research work, the vibration signal is required to be classified into four classes therefore, the binary SVM is extended to multi-class SVM via a one-versus-all strategy. Interested readers are referred to [55,56], for proper discussion on SVM. Here, the Lib-SVM [57] library is used to compute the parameters of the hyper-plane. It uses the following objective function with the training data provided, to optimise the hyper-plane using the training data:(3)$$\begin{array}{cc} \; & \min_{w,b,\xi }\left(\frac{1}{2}w^{⊤}w+C\sum_{j}\xi_{j}\right)\\ \; & s.t.\;\;y_{j}\left(ws_{j}+b\right)\geq 1-\xi_{j},\xi_{j}\geq 0, \end{array}$$
where *C* is the regularization constant, the hyperplane is represented by w and *b*, while the non-separable cases are incorporated by $\xi^{j}$. For non-linear separation, the constraint $y_{j}\left(w\phi \left(s_{j}\right)+b\right)\geq 1-\xi_{j},\xi_{j}\geq 0$ can be introduced to perform the computation in an implicit higher dimensional space. The label $y_{t}$ of a test feature vector $s_{t}$ is determined using the sign of $\frac{ws_{t}+b}{∥w∥}$, once the parameters of the optimal hyper-plane are calculated.

KLDA uses supervised dimensionality reduction to represent data more efficiently, suppressing the less useful features for classification. It is applied to non-linearly separable classes that transform the *p* dimensional training feature vectors to c−1 dimensional vectors which are then used for classification using machine learning techniques like SVM and so forth. KLDA uses Kernel matrix, computed using the function: $K\left(i,j\right)=k\left(s_{i},s_{j}\right)$. Given an input kernel K, KLDA solves the following objective function [58]:(4)$$\alpha_{opt}=\arg\max\frac{\alpha^{⊤}KWK\alpha }{\alpha^{⊤}KK\alpha },$$
where $\alpha ={\left[\alpha_{1},...,\alpha_{g}\right]}^{⊤}$. $W\in R^{g\times g}$ is a block-diagonal matrix: $W=diag\{W_{1},W_{2},...,W_{c}\}$, where $W_{j}\in R^{m_{j}\times m_{j}}$ have every elements equal to $\frac{1}{m_{j}}$ ($m_{j}$ represent the number of samples in class *j*). The largest eigenvectors of ${\left(KK+\epsilon I\right)}^{-1}\left(KWK\right)\alpha =\lambda \alpha$ gives the optimal solution. The (c−1) dominant eigenvectors ($\Lambda =\left[\alpha_{1},...,\alpha_{c-1}\right]\in R^{p\times (c-1)}$) are used to construct the transformation matrix and the training data matrix is projected on Λ to perform dimensionality reduction.

The Average, Kurtosis, Skewness and Standard Deviation vectors of each domain were concatenated before giving to SVM, KNN and KLDA for classification into bearings with outer race fault, inner race fault, and ball fault. The results are shown in Table 1. The first column shows the feature vector along with its dimensions. In the first column Statistical_P, Fourier_P and PSD_P show our proposed feature vectors. The Statistical_P is a concatenation of Average, Kurtosis, Skewness and Standard Deviation of the vibration signal along with the same features calculated after taking the second difference of the vibration signals. Thus making a total of eight feature vectors of a size of two hundred and twenty eight each. The vectors of Fourier_P and PSD_P were arranged in a similar way, while the EMD, Fourier and PSD were calculated without taking the Average, Kurtosis, Skewness, and Standard Deviation and were used for comparison. The highest accuracy is given by the PSD_P for KLDA, which is 99.13%, followed by Statistical_P, which is 98.257% using KLDA. The Fourier is third in the row with 98.0% using KLDA followed by EMD with 97.01% using KNN. It is clear that our proposed feature vectors outperformed some of the most widely used feature vectors referenced in the literature. To further authenticate the performance of the proposed method, a reduction percentage is presented in Table 2. It can be observed that with the proposed method, more than 95% reduction level is achieved, which clearly decreases the computational complexity and also the classification time.

### Statistical Significance

Models are mostly evaluated by utilising the resampling methods such as k-fold cross-validation or hold-out from which the mean scores are determined and later compared. Though, the approaches are simple and valid, but could be misleading as it is quite hard to recognise that the difference between the scores is real or the results are statistically not stable. Therefore, statistical tests offer this confidence by quantifying the likelihood of the samples-following the same distribution. The main motivation behind this statistical analysis is to acquire an absolute level of confidence in the implemented scheme. In this work, we utilise the analysis of variance (ANOVA) [59] statistical model to validate the significance of accuracy intervals—simply by analysing the selected samples for the differences among means.

For the proposed scenario, we are considering two different classifiers ($KNN^{q}$ & $KLDA^{r}$)—selected on the basis of their improved classification results. In ANOVA, multiple tests are performed for the presumption of homogeneity of variance and normality. A Bartlett’s test is performed to verify the homogeneity of variance, whereas the Shapiro–Wilk test is performed to check normality. The significance level α is selected to be 0.05, which indicates if the test value found greater than α then data would be normally distributed and vice versa. The means for the given samples are represented by $\bar{x_{1}},\bar{x_{2}}$, which are calculated by performing the Monte-Carlo simulations to identify the lower and upper bounds after simulated 500 times. The null hypothesis $H_{0}$ claims that for the given means $\bar{x_{1}}=\bar{x_{2}}$, whereas the alternative hypothesis $H_{a}$ claims the rejection, $\bar{x_{1}}\neq \bar{x_{2}}$. To test the null hypothesis, $H_{0}$, *p*-value is calculated—conceded that the rejection fulfils the relation, p≤σ, otherwise, the Bonferroni post-hoc test will be performed [60].

In this work, three different scenarios are considered, which are the calculation of statistical parameters on; (1) time-series data, (2) Fourier descriptors, and (3) PSD. In the first case (time-series data), by using the selected classifiers ($KNN^{q}$, & $KLDA^{r}$), the Shapiro–Wilk test generated the *p*-values, $p^{q}=0.3203$, and $p^{r}=0.9792$. Similarly, by applying Bartlett’s probability test, the associated Chi-squared probabilities are: $c^{q}=2.277$, and $c^{r}=0.420$. For the second case (Fourier descriptors), the Shapiro–Wilk test generated the *p*-values, $p^{q}=0.9206$, and $p^{r}=0.9724$. Similarly, by applying Bartlett’s probability test, the associated Chi-squared probabilities are $c^{q}=0.1668$, and $c^{r}=0.0698$. Finally, for the last case (PSD), the associated *p*-values are $p^{q}=0.7386$, and $p^{r}=0.1616$. By performing the Bartlett’s probability test, the calculated probabilities are $c^{q}=0.606$, and $c^{r}=3.645$.

In can be observed from the calculated *p*-values, considering all the cases and using the selected classifiers, are greater than σ. Thus, from the both probability test results (equality of variances, and normality), we fail to reject the null hypothesis $H_{0}$, and therefore certain about the claim that the test data is normally distributed, and with the homogeneous variances. In Table 3, Table 4 and Table 5, we present a few statistical parameters to verify the authenticity of the proposed method based on the classification results. The ANOVA test parameters include degree of freedom (df), sum of squared deviation (SS), mean squared error (MSE), F-statistics and Prob-F value. Similarly, the confidence interval of all the selected classifiers and the selected cases are depicted in Figure 7, Figure 8 and Figure 9. where the horizontal label 1 represents KNN classifiers, and 2 represents KLDA classifier.

The classification time of the proposed method is also provided in Figure 10, where it can be observed that the maximum time to classify any feature set is 12.62 s after achieving the reduction percentage of more than 95%. Whereas, Figure 11 depicts the classification time of original feature vectors in minutes. One can observe the time taken using the EMD features, which is more than 80 min, in comparison to PSD and Fourier, which are 11.34 min and 13.65 min, respectively. The classification time comparison is also provided in Figure 10.

## 5. Conclusions

In this work, the vibration signals were analysed to detect and classify faults in rotating machinery. The signal was recorded and its statistical features, such as Average, Kurtosis, Skewness and RMS, were calculated in the time domain and the frequency domain. These features were also calculated by first finding the second derivative of the raw time domain signal. The features were then fed to different machine learning algorithms and were analysed for different patterns due to different faults and were used to train these machine learning models, resulting in successful detection and classification into ball, inner race and outer race faults. The Power Spectral Density features showed the best results for KLDA, followed by the statistical features using KLDA. This result was compared with that of the EMD, Fourier Transform and Power Spectral Density, in which the former one is time-frequency while the latter two are frequency domain representation. It is also important to note that the sizes of our proposed features are much less than those of the EMD, Fourier and Power Spectral Density, showing the computational efficiency of our proposed techniques. The proposed technique can be extended to time-frequency analyses like Short Term Fourier Transform and Wavelet Transform and so forth; also other bearing faults can be added, such as cage fault, which is not addressed here.

## Figures and Tables

**Figure 1 sensors-22-02012-f001:**
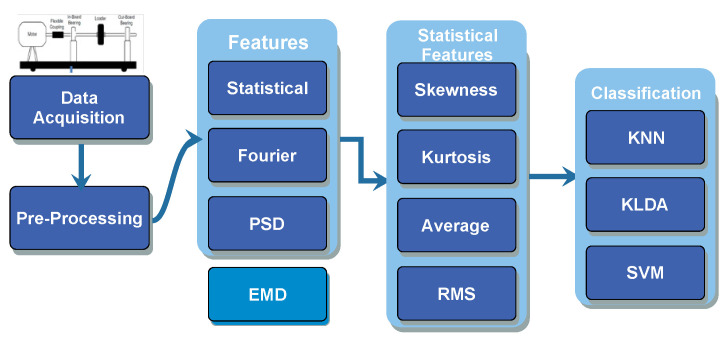
Block Diagram of Proposed Procedure.

**Figure 2 sensors-22-02012-f002:**
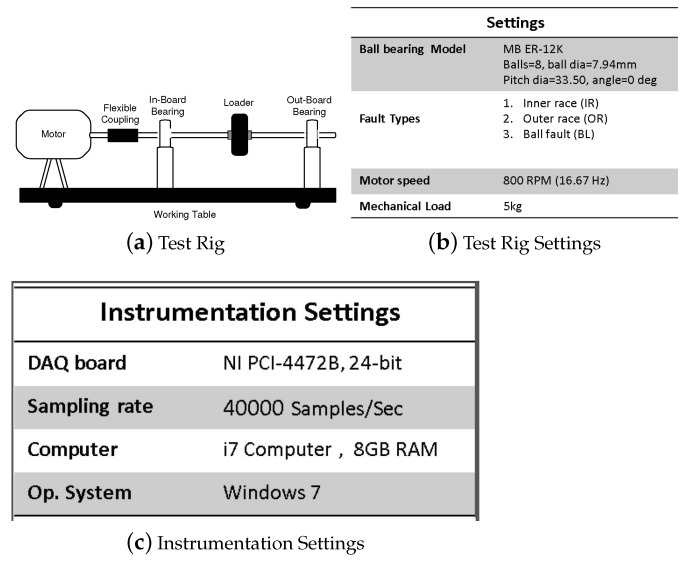
Test rig block diagram and specifications.

**Figure 3 sensors-22-02012-f003:**
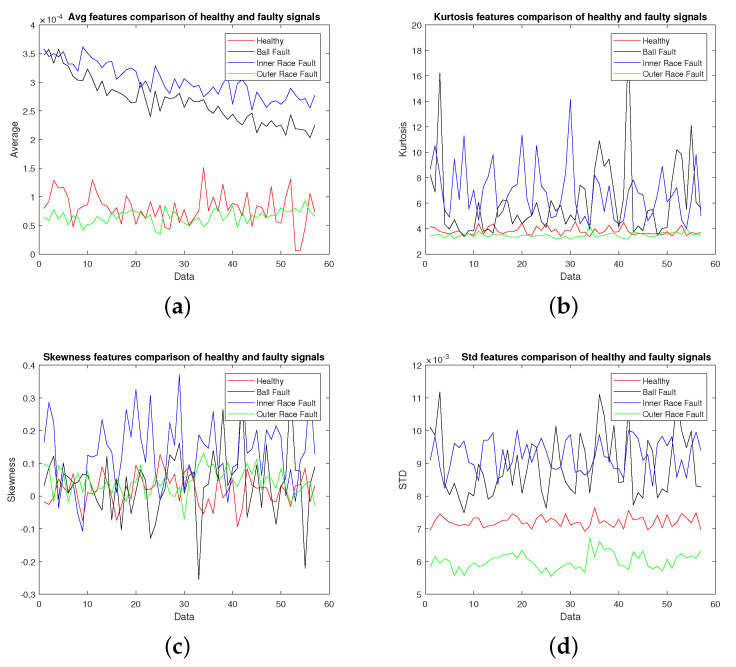
Statistical signal demonstration: (**a**) Average, (**b**) Kurtosis, (**c**) Skewness, (**d**) Standard Deviation.

**Figure 4 sensors-22-02012-f004:**
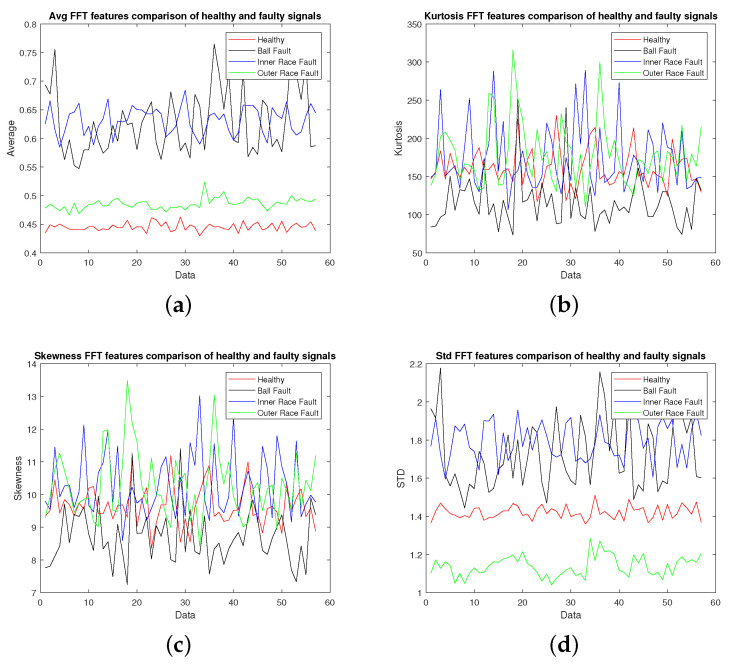
FFT signal demonstration: (**a**) Average, (**b**) Kurtosis, (**c**) Skewness, (**d**) Standard Deviation.

**Figure 5 sensors-22-02012-f005:**
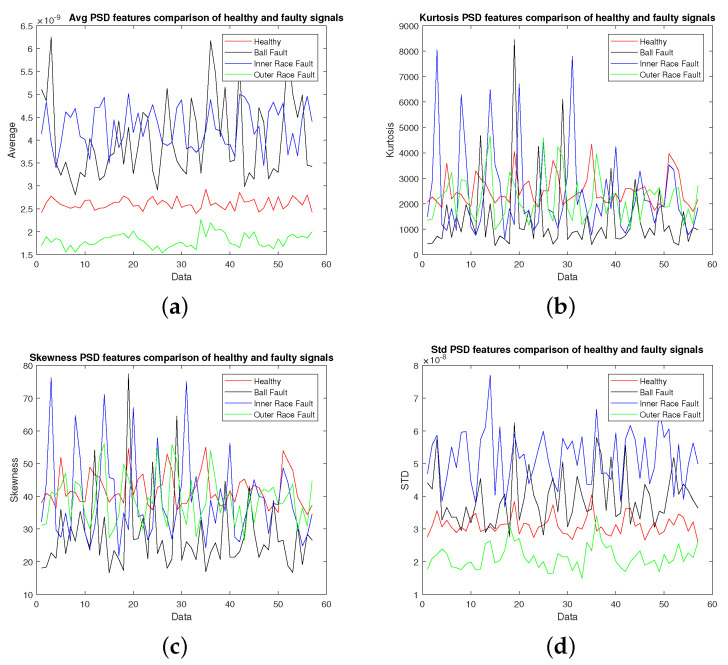
PSD signal demonstration: (**a**) Average, (**b**) Kurtosis, (**c**) Skewness, (**d**) Standard Deviation.

**Figure 6 sensors-22-02012-f006:**
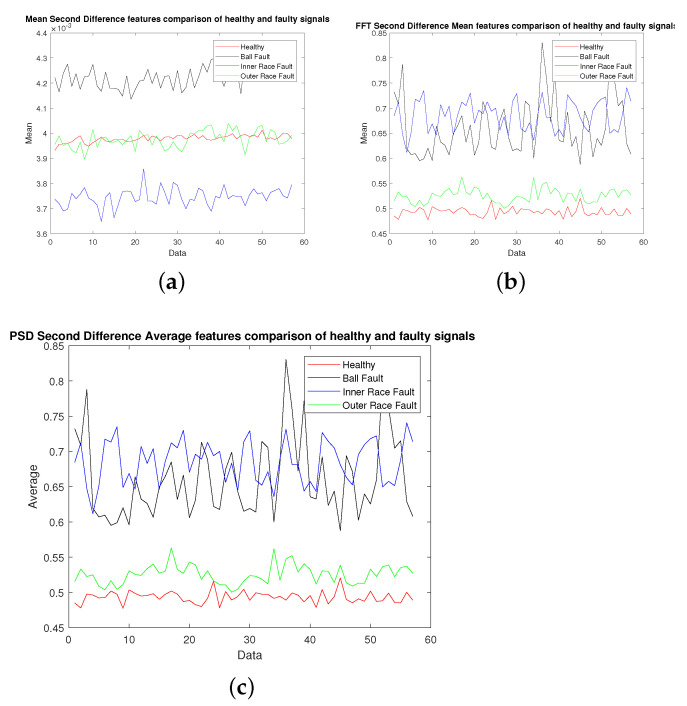
Analysis of mean values of second derivative: (**a**) Average (Time Domain), (**b**) Average FFT, (**c**) Average PSD.

**Figure 7 sensors-22-02012-f007:**
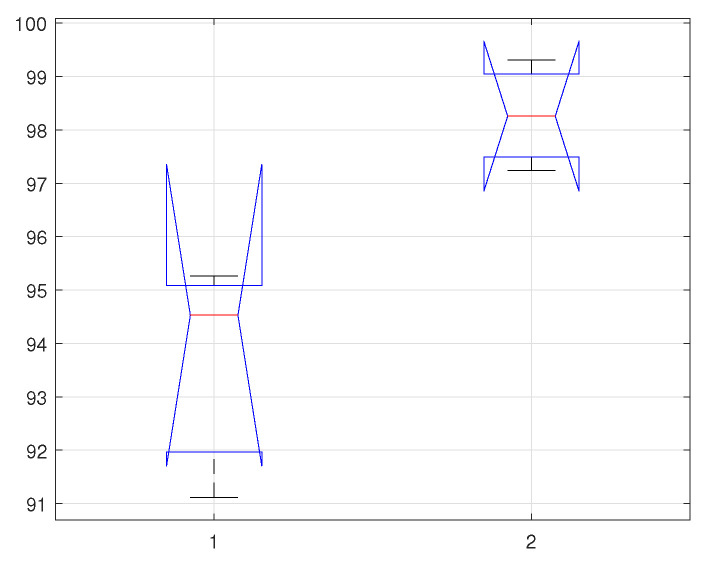
Confidence Interval of selected classifiers using statistical features of time-series data.

**Figure 8 sensors-22-02012-f008:**
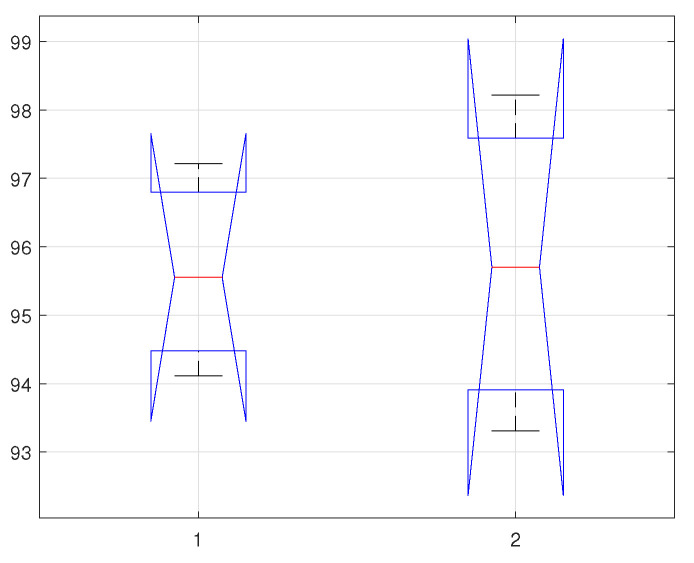
Confidence Interval of selected classifiers using statistical features of fourier descriptors.

**Figure 9 sensors-22-02012-f009:**
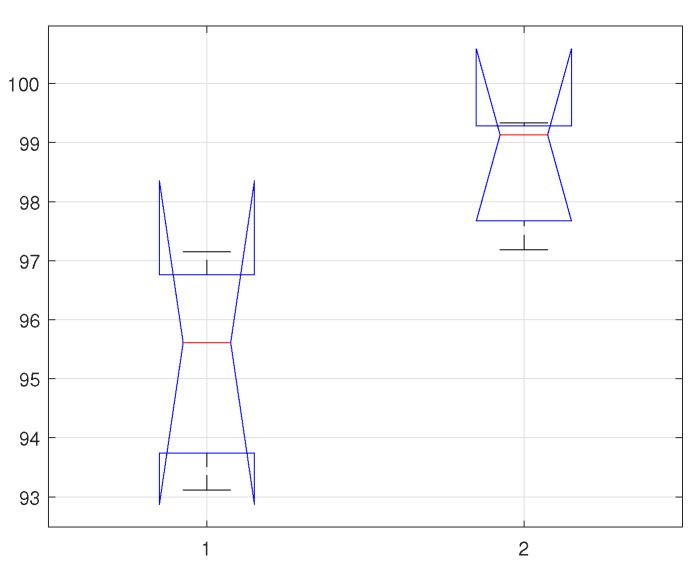
Confidence Interval of selected classifiers using statistical features of PSD data.

**Figure 10 sensors-22-02012-f010:**
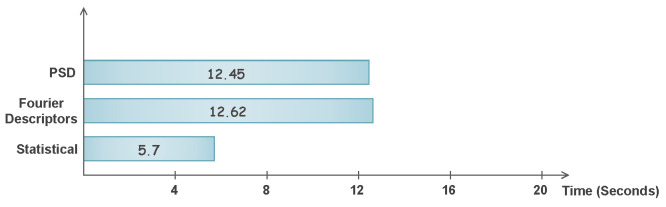
Classification time of three feature sets using proposed technique in seconds.

**Figure 11 sensors-22-02012-f011:**
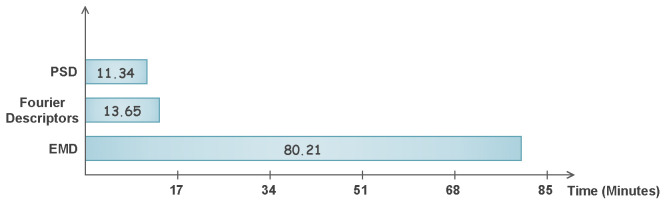
Classification time of three feature sets using proposed technique in minutes.

**Table 1 sensors-22-02012-t001:** Classification of faults.

Classifier	SVM	KNN	KLDA
Statistical_P	66.11%	93.53%	98.27%
Fourier_P	59.16%	95.52%	95.70%
PSD_P	62.25%	95.64%	99.13%
EMD	80.49%	93.01%	66.06%
Fourier	60.71%	94.13%	97.04%
PSD	65.65%	94.99%	85.47%

**Table 2 sensors-22-02012-t002:** Data dimensions and their reduction percentage.

Original Data Vector	Feature Vector	Final Feature Vector	Reduction Percentage
1 × 2,420,000	Statistical (8 × 228)	(8 × 228)	99.924
	Fourier (1 × 40,000)	(8 × 228)	95.44
	PSD (1 × 40,000)	(8 × 228)	95.44
	EMD (14 × 160,000)	(14 × 160,000)	7.438

**Table 3 sensors-22-02012-t003:** ANOVA test on statistical time-series data based on the selected classifiers.

Variance Source	SS	df	MS	F	Prob > F
Columns	32.1785	1	32.1785		0.0304
Error	11.9377	4	2.9844	-	-
Total	44.1162	5			

**Table 4 sensors-22-02012-t004:** ANOVA test on statistical Fourier data based on the selected classifiers.

Variance Source	SS	df	MS	F	Prob > F
Columns	0.0201	1	0.02007	0.001	0.9483
Error	16.8298	4	4.20746	-	-
Total	16.8499	5			

**Table 5 sensors-22-02012-t005:** ANOVA test on statistical PSD data based on the selected classifiers.

Variance Source	SS	df	MS	F	Prob > F
Columns	15.9218	1	15.9218	5.75	0.0746
Error	11.0826	4	2.7706	-	-
Total	27.0044	5			

## Data Availability

Not Applicable.

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
