# Peer review of "A New Statistical Features Based Approach for Bearing Fault Diagnosis Using Vibration Signals"

_sensors, 2022, doi:10.3390/s22052012_

Round 1
Reviewer 1 Report
there are publications with similar classification accuracy with only time domain signals. What is the need for tranforming into frequency domain and waste resources and time?
Author Response
The response sheet has been attached. thanks

Reviewer 2 Report
sensors-1496829
A New Statistical Features Based Approach for Bearing Fault Diagnosis Using Vibration Signals
Authors claimed that they investigated several statistical indexes for bearing fault classification usage. Experimental works are performed to check the performance of the method. Generally the novelty of the manuscript is okay. However, some critical comments are also listed below.
- The introduction is not written well for the statistical feature-based method have been proposed before, see, DOI10.1016/j.ymssp.2018.04.038 and DOI10.1155/2021/5554316.
- Please give more details about the test rig and measurement systems to collect raw signals. By the way, the pictures of faulty components should also be given.
- Why not use time-frequency domain indexes?. In our research, the indexes might be useful for bearing fault detection.
- There are several spelling mistakes in the text that need to be corrected, for example, ‘be performed [? ].’…
Author Response
Response sheet has been attached. thanks

Reviewer 3 Report
Dear Authors,
Based on the first round review of the manuscript entitled A New Statistical Features Based Approach for Bearing Fault Diagnosis Using Vibration Signals, the reviewer has the following comments:
- Please more explain about the contributions in the revised manuscript.
- Why the authors extract the features from the 2nd derivative of the signals? it means the 2nd derivative of the acceleration?
- How you can validate the reliability and stability of the proposed technique? please explain about it in the revised manuscript.
- Based on the block diagram of the proposed method, (Fig.1), this approach has 4 parts including preprocessing, features, statistical feature, and classification. Please explain in which part/parts you have contribution? please explain deeply in the revised manuscript.
- Please explain about the dataset in the separate section and explain deeply in the revised manuscript.
- How you can validate the robustness of the proposed method? please explain in the revised manuscript.
- What is the limitation of the proposed method? please mention about it in the conclusion.
- What is the future work? please explain about it in the conclusion.
Regards,
Author Response

(The authors gave the same response as above.)

Round 2
Reviewer 1 Report
The paper does not add any value to the scientific community. The author response to my comment is not satisfactory.
Reviewer 2 Report
no further comments.
Reviewer 3 Report
Dear Authors,
Thank you for your response letter. Regarding the 2nd round review of the manuscript entitled: A New Statistical Features Based Approach for Bearing Fault Diagnosis Using Vibration Signals, it can be accepted for further processing.
Regards,